# Health and well-being issues of Nepalese migrant workers in the Gulf Cooperation Council countries and Malaysia: a systematic review

Priyamvada Paudyal [iD],[1] Kavian Kulasabanathan,[1] Jackie A Cassell [iD],[1] Anjum Memon [iD],[1] Padam Simkhada,[2] Sharada Prasad Wasti[3]

► Prepublication history and supplemental material for this paper are available online. To view these files, please visit the journal online (http://dx.doi.org/10.1136/bmjopen-2020-038439).

[1]Department of Primary Care and Public Health, Brighton and Sussex Medical School, Brighton, UK
[2]Department of Allied Health Professions, Sport and Exercise, Faculty of Health, University of Huddersfield, Huddersfield, UK
[3]Green Tara Nepal, Kathmandu, Nepal

**Correspondence to**
Dr Priyamvada Paudyal;
p.paudyal@bsms.ac.uk

## ABSTRACT

**Objective** To summarise the evidence on health and well-being of Nepalese migrant workers in the Gulf Cooperation Council (GCC) countries and Malaysia.

**Design** Systematic review.

**Data sources** EMBASE, MEDLINE, Scopus and Global Health databases.

**Eligibility criteria** Studies were eligible if they: (1) included Nepalese migrant workers aged 18 or older working in the GCC countries or Malaysia or returnee migrant workers from these countries; (2) were primary studies that investigated health and well-being status/issues; and (3) were published in English language before 8 May 2020.

**Study appraisal** All included studies were critically appraised using Joanna Briggs Institute study specific tools.

**Results** A total of 33 studies were eligible for inclusion; 12 studies were conducted in Qatar, 8 in Malaysia, 9 in Nepal, 2 in Saudi Arabia and 1 each in UAE and Kuwait. In majority of the studies, there was a lack of disaggregated data on demographic characteristics of Nepalese migrant workers. Nearly half of the studies (n=16) scored as 'high' quality and the rest (n=17) as 'moderate' quality. Five key health and well-being related issues were identified in this population: (1) occupational hazards; (2) sexual health; (3) mental health; (4) healthcare access and (5) infectious diseases.

**Conclusion** To our knowledge, this is the most comprehensive review of the health and well-being of Nepalese migrant workers in the GCC countries and Malaysia. This review highlights an urgent need to identify and implement policies and practices across Nepal and destination countries to protect the health and well-being of migrant workers.

## Strengths and limitations of this study

► This review is the most comprehensive review to date on this population.
► The review did not restrict studies based on particular health outcomes, peer-reviewed studies looking at a range of health issues in this population were included.
► Meta-analysis was not conducted as there was heterogeneity in the outcome measured and the measurement tools used in the studies.

achieve Universal Health Coverage for all.[1] According to the World Migration Report 2020, the number of international migrants has reached approximately 272 million, and two-thirds of these are estimated to be labour migrants.[2] Labour migration has been a key determinant of population changes in Asia, especially in Gulf Cooperation Council (GCC) countries, a major destination for workers within Asia.[2]

Nepal is a low-income country going through a demographic transition, with an ageing population and attendant chronic diseases. According to the Nepal Migration Report 2020, over 4 million labour approval were issued to Nepalese workers in the last decade since 2008/2009.[3] The Nepal Demographic and Health Survey (2016) reported that nearly half (47%) of the households have at least one family member who migrated in the last 10 years either in internal or international destinations.[4] These migrant workers contribute over a quarter of the country's gross domestic product through remittance from abroad. The migration outflow consists predominantly of low-skilled male workers, primarily to Malaysia and the GCC countries.[3]

Labour migration contributes significantly to the sociocultural and economic development of both origin and destination

## INTRODUCTION

Migration is the overarching narrative of our time, and its impact is increasingly being recognised in global public health agendas. The United Nations (UN) sustainable development goals identify migration as a catalyst for development and recommend that 'no-one should be left behind' to

countries. However, migrant workers experience specific vulnerabilities, and face a range of health risks while working abroad. These risks are particularly significant for Nepalese workers in the GCC countries, as they are often employed in occupations considered 'difficult, dirty and dangerous' (3Ds). These are sectors with higher occupational risks such as agriculture, construction, transport and heavy industry. Furthermore, Nepalese migrant workers consistently work for longer hours as compared with native workers[5 6] and are often exposed to factors which promote poor health and well-being, including low wages, poor housing, an unhealthy diet and difficulty in accessing health services.[5 7] Many Nepalese migrant workers die abroad every year including a significant number that are unexplained, while a large number return home with debilitating injuries, and both mental and physical illness.[5] This systematic review identified and summarised the evidence from primary studies on the health and well-being of Nepalese migrant workers in the GCC countries and Malaysia, the destination countries for 88% of labour migration. This review was conducted as a part of University of Sussex internally funded Global Challenges Research Fund project to develop a culturally relevant intervention to support the health and well-being of Nepalese migrant workers in GCC countries.

## METHODS
### Protocol registration
This study protocol was registered at the University of Sussex (http://sro.sussex.ac.uk/id/eprint/86400/). The study followed the PRISMA (Preferred Reporting Items for Systematic Reviews and Meta-Analyses) guidelines and recommendations of the Cochrane Collaboration (www.prisma-statement.org).

### Electronic search
A combination of migration specific search terms (migration, migrant, emigrant, immigrant, expatriate, foreign worker, labour migration, left-behind, migrant families) and country specific search terms (Nepal, Nepalese, Nepali, UAE, United Arab Emirates, Gulf Cooperation Council (GCC), Middle East, Bahrain, Saudi Arabia, Oman, Qatar, Kuwait, Malaysia) were used to identify relevant studies using EMBASE, MEDLINE, Scopus and Global Health databases (see online supplemental appendix 1). The search aimed to identify all relevant studies regardless of any health outcomes used. As such, no health outcome specific terms were used to limit the electronic search. Reference lists of the relevant studies including those of related systematic reviews and reference lists of the selected studies were further screened to identify potentially eligible studies.

### Inclusion and exclusion criteria
Studies were eligible if they: (1) included Nepalese migrant workers aged 18 or older working in the GCC countries or Malaysia or returnee migrant workers from

these countries; (2) provided primary data on health and well-being status/issues (physical health, mental health, accidents and injuries); and (3) were published in English language before 8 May 2020.

### Article screening and selection
Once the electronic search was completed, the identified articles were exported to Rayyan (https://rayyan.qcri.org/welcome) and screening was carried out by two reviewers (SPW and KK) independently to identify eligible articles. The titles of the identified studies were screened to remove any duplicates and irrelevant articles. The abstract of all remaining articles was screened to identify eligible full text articles. Full text articles were reviewed and a consensus was reached to finalise the articles for inclusion. If more than one study were published using the same data source (eg, routine healthcare date), we used the study with the largest sample size. Any disagreement over eligibility of studies was resolved through discussion with the third reviewer (PP).

### Data extraction and synthesis
The information extracted from each article included: study reference (authors, publication year and country), study design and settings, participants' characteristics (sample size, age and gender), health outcomes and key findings (table 1). Extracted data were analysed and a summary of the narrative synthesis is reported in the results section. Meta-analysis was not conducted as there was heterogeneity in the outcome measured and the measurement tools used in the studies.

### Quality assessment
The PRISMA guideline suggests that systematic review should assess the risk of bias (based on theoretical grounds) rather than study quality (the best authors could do in the setting). However, we assessed the latter as the studies included in this review were predominately cross-sectional in nature with methodological limitations.[8] Quality assessment for this review was done using the Joanna Briggs Institute (JBI) Critical Appraisal Tools.[9] The JBI prevalence study critical appraisal tool was used for cross-sectional studies estimating the prevalence of the condition. The tool contains nine items covering domains related to sampling, outcome assessment, statistical analysis and response rate. Each item was scored one if the response was 'Yes' and scored zero if the response was 'No' or 'Unclear'. As in the previous review,[10] studies with eight or more 'Yes' response were rated as 'high' quality, four to seven as 'moderate' and three or below as 'low' quality. Similarly, the JBI analytical cross-sectional study critical appraisal tool was used for cross-sectional studies reporting effect sizes. The checklist contains eight items covering domains related to sampling, exposure, outcome, confounding factors, and statistical analysis (maximum possible score eight). Studies were categorised as high quality (seven or above), moderate quality (between five and six) or low quality

**Table 1** Characteristics of studies included (n=33)

| Author, country | Study design and setting | Participant characteristics | Health outcomes (measurement tools) | Key findings | QA scores |
|---|---|---|---|---|---|
| **Health risk and experiences related issues** | | | | | |
| Dhakal et al,[37] Nepal | Hospital record data evaluated from the hospital data in Nepal (January–July 2019) | Returnees migrant participants– 44 (n=42) Gender—Male – 95% Age—Mean age 37.2 years | Healthcare access and prevalence of chronic kidney disease (CKD) | ▶ Workers with health insurance 68.2% (95% CI 52.4 to 81.3) (n=30)<br>▶ Underwent for routine health check-ups annually 20.4% (95% CI 9.8 to 35.3) (n=9)<br>▶ No regular health check-up 79.5% (95% CI 64.7 to 90.0) (n=35)<br>▶ Exposed to chemicals 27.3% (95% CI 14.9 to 42.7) (n=12)<br>▶ Patients were unknown about cause of CKD 77.3% (95% CI 62.1 to 88.5) (n=34)<br>▶ Had diabetic nephropathy 13.6% (95% CI 5.1 to 27.3) (n=6)<br>▶ Death due to kidney failure (n=1) | Moderate |
| Khaled and Gray,[21] Qatar | Cross-sectional survey, February 2016 | Migrant workers in Qatar Total participants—2520 Nepalese—26% (n=655) Gender—NR Age—NR | Depressive symptoms | ▶ Compared with Arabs, Nepalese migrant experienced 4%, increase in the predicted probability of depressive symptoms, for every unit increase in perceived quality of life | Moderate |
| Regmi et al,[38] Nepal | Qualitative Study (data collected in 2017) | Returnee migrants in Nepal from Qatar, Saudi Arab, Malaysia, Oman, UAE Sampled—20 | Various health issues | ▶ Unfair treatment and discrimination at work<br>▶ Poor working and living arrangements—dirty toilets and bathrooms<br>▶ Lack of security, loneliness and poor social life at work place/ social isolation<br>▶ Mental health problems—tensions, anxiety and attempt to suicide and poor access to mental health services<br>▶ Poor communication facilities<br>▶ Only formality of pre-departure training package—contents good but poor implementation | Moderate |
| Adhikary et al,[36] Nepal | Qualitative study (July–September 2011) | Returnee migrants, interviews conducted interviews in Nepal—20 Male—all Mean age—31.3 years | Workplace accidents among Nepali male workers in Qatar, Saudi Arabia and Malaysia | ▶ Work place related issues:<br>▶ Not safe workplace<br>▶ High work pressure<br>▶ No medical supports from employer in host country<br>▶ Long working hours, mostly without timely food and drinking water resulting in dehydration and heat stroke<br>▶ Communication difficulty due to language barriers<br>▶ Injuries and accidents related issues<br>▶ Fall from the roof, trapped in the hole<br>▶ Injured back bone, legs, hands and head<br>▶ Life-long disability | High |

**Table 1** Continued

| Author, country | Study design and setting | Participant characteristics | Health outcomes (measurement tools) | Key findings | QA scores |
|---|---|---|---|---|---|
| Pradhan et al,[35] Nepal | Retrospective analysis of Government of Nepal provided data (2009–2017) | Nepali migrant workers in Qatar Total sample—1354 Gender—NR Age—NR | Analysed the deaths of Nepalese migrant workers | ▶ Causes of death due to: ▶ Cardiovascular—42% (95% CI 39.5 to 42.8) (n=571) ▶ Suicide—8.5% (95% CI 7.1 to 10.1) (n=116) ▶ Workplace accident—12.4% (95% CI 10.7 to 14.3) (n=169) ▶ Road traffic accident—10.1% (95% CI 8.5 to 11.8) (n=137) ▶ Murder—1.7% (95% CI 1.0 to 2.5) (n=23) ▶ Natural/others reasons for death—25% (95% CI 22.6 to 27.3) (n=338) | Moderate |
| Adhikary et al,[33] Nepal | Cross-sectional questionnaire-based survey | Male Nepalese construction workers, worked in host countries (Malaysia, Qatar and Saudi Arabia) for >6 months. Total participants—403 Age—NR | Self-reported health and well-being status | ▶ 13.2% (95% CI 10.0 to 16.8) (n=53) reported poor/very poor health, relating to: ▶ Age older than 40 years reported as poor health (OR=3.0, 95% CI 1.0–9.0) ▶ Poor work environment (OR=6.8, 95% CI 3.2 to 14.6) ▶ Health risks at work (OR=4.7, 95% CI 2.1 to 10.5) ▶ Prevalence of mental health issues was 23% overall—strong link between perceived health risks and mental health status | Moderate |
| Adhikary et al,[32] Nepal | Cross-sectional questionnaire-based survey | Male Nepalese construction and factory workers, worked >6 months in Malaysia, Qatar or Saudi Arabia. Total participants—423 Age—NR | Self-reported perceived health risks and accidents at work | ▶ Poor or very poor work environment (rated by the workers) associated with greater perceived health risk at work (OR 2.5, 95% CI 1.5 to 4.4) ▶ Prevalence of accidents at work=17% ▶ Variables associated with accidents at work included: >age 40 and above vs 20–29 (OR=4.0, 95% CI 1.7 to 9.7) ▶ Not satisfied accommodation vs satisfied with accommodation (OR=1.9, 95% CI 1.1 to 3.4) ▶ Poor or very poor work environment vs good/good to fair environment (OR 3.5, 95% CI 1.8 to 6.7) ▶ Working in Middle-East vs Malaysia (OR .3.6, 95% CI 1.5 to 8.5) ▶ Not registered with a doctor vs registered (OR=0.3, 95% CI 0.1 to 0.7) | Moderate |
| Simkhada et al,[57], Nepal (data for GCCs and Malaysia provided by the authors) | Retrospective analysis of NGO collected data (July 2009– July 2014) via Paurakhi Nepal (NGO) | Returnee Nepalese female migrant workers from GCC and Malaysia Total participants—942 GCC=933 Malaysia=9 Median age 31 (IQR 37) Age range—14–51 years | Various health issues while working in GCC, middle-east and Malaysia (prevalence calculated using information available from client Information Form/Sheet) | ▶ Proportion women with health problems—24% (95% CI 21.3 to 26.8) (n=226) ▶ Abuse at workplace—37% (95% CI 33.6 to 39.9) (n=346) ▶ Accident at workplace—1.1% (95% CI 0.5 to 1.2) (n=10) ▶ Mental health problem—8.3% (95% CI 6.6 to 10) (n=78) ▶ Torture or maltreatment at the workplace 30.9% (95% CI 27.9 to 33.9) (n=291) ▶ Pregnancy at work place—3.1% (95% CI 2.1 to 4.3) (n=29) ▶ Sexual abuse—51.7% (95% CI 32.5 to 70.5) (n=15/29) ▶ Physical harm—10.9% (95% CI 9.0 to 13.1) (n=103) ▶ Received health services—10.8% (95% CI 8.9 to 12.9) (n=102) | Moderate |

Continued

**Table 1** Continued

| Author, country | Study design and setting | Participant characteristics | Health outcomes (measurement tools) | Key findings | QA scores |
|---|---|---|---|---|---|
| Irfan et al,[22] Qatar | Cross-Sectional study (June 2012–May 2013) | Patients attending to the emergency medical service in Qatar Total participant—447 Nepalese—11.6% (n=52) Gender—NR Age—median age 51 years (range 39–66 years) | Proportion of out of hospital cardiac arrest | ▲ Out-of-hospital cardiac arrest among Nepalese migrant patients—11.6% (95% CI 8.8 to 14.9) (n=52). No further data | High |
| Min et al,[25] Malaysia | Retrospective cross-section of routine healthcare data (January 2011–December 2013) | Patients attending to the eye casualty with work-related ocular injuries, in Hospital Sultan Ismail in Johor Bahru, Malaysia Total 440 work-related ocular traumas. Nepalese—21.7% (n=33) Gender—NR Age—NR | Work related ocular traumas | ▲ 33 cases of Nepalese work-related eye injuries. Causes range from open globe injuries due to being hit by a machine, nail, wood and metal while grinding | High |
| Al-Thani et al,[15] Qatar | Retrospective analysis of hospital trauma registry records 2010–2013 Hamad Trauma Centre | Total migrant participants—2015 Nepalese—28% (n=563) Male—98% (n=1972) Female—2% (n=43) Age—NR | Proportion of occupational injuries and mortality cases | ▲ Overall proportion of occupational injury cases—27.9% (n=563), of which ▲ Falls from height—52.4% (95% CI 48.1 to 56.5) (n=295) ▲ Fall of a heavy object—20.4% (95% CI 17.1 to 24) (n=115) ▲ Motor vehicle crashes injuries—17% (95% CI 14.2 to 20.6) (n=97) ▲ Machinery injuries—5% (95% CI 3.1 to 6.9) (n=27) ▲ Others—5% (95% CI 3.4 to 7.3) (n=29) | Moderate |
| Latifi et al,[20] Qatar | Retrospective analysis of routine healthcare data | Total traffic related pedestrian injuries (TRPI) patients—601 Total Nepalese expat TRPI patients—(n=147) Gender—NR Age—NR | Pedestrian morbidity and mortality | ▲ 25.4% (95% CI 21.0 to 18.0) of TRPI were of Nepalese migrant workers (vs 16.0% of the general population of Qatar being Nepalese) ▲ 51.4% of TRPI with positive blood alcohol were Nepalese migrant workers | High |

Continued

**Table 1** Continued

| Author, country | Study design and setting | Participant characteristics | Health outcomes (measurement tools) | Key findings | QA scores |
|---|---|---|---|---|---|
| Joshi et al,[34] Nepal | Cross-sectional study | Nepalese migrants with experience of >6 months in Qatar, Saudi Arabia or United Arab Emirates. Total participants—408 Males—92.4% (n=377) Aged between 26 and 35—53.4% (n=218) | Knowledge of HIV/AIDS and risk perceptions | ▶ Risk perceptions of HIV/AIDS: ▶ Concerned about HIV/AIDS—90% (95% CI 86.3 to 92.4) (n=366) ▶ Perceived themselves at high risk of being infected due to their sexual activities—59.2% (n=397) ▶ Sexual behaviour: ▶ 17.2% (95% CI 13.6 to 21.1) (n=70) had sexual intercourse with a partner other than their spouse during the last 12 months of their stay abroad | Moderate |
| Kavarodi et al,[19] Qatar | Population-based cross-sectional study | Low income expatriate workers from Indian sub-continent (living in Qatar for >6 months) Total participants—3946 Nepalese—5.4% (n=213) Gender—NR Age—NR | Clinical prevalence of suspected oral lesions | ▶ Oral lesions in of Nepalese workers 4.7% (95% CI 2.1 to 7.8) (n=10) | High |
| Alswaidi et al,[39] Saudi Arabia | Review of Ministry and Health data from Saudi expat worker fitness screening programme (1997–2010) | Total number of registered expatriate workers—4 272 480 Nepalese—0.9% (n=38 908). Females—14% (n=5367) Males—86% (n=33 541) Age—NR | Proportion of 'unfit' to workers | ▶ Cases of unfitness among Nepalese workers by gender: ▶ Unfit men—1.99% (95% CI 1.8 to 2.1) (n=669) ▶ Unfit women—1.2% (95% CI 0.9 to 1.5) (n=64) ▶ Overall unfit—1.9% (95% CI 1.7 to 2.0) (n=733) ▶ Nepalese migrants were the third most unfit population ▶ Nepalese migrants as proportion of all those with: ▶ Infectious causes of unfitness (incl. hepatitis, HIV, tuberculosis (TB))—1.6% (n=379) ▶ Non-communicable causes of unfitness—5.3% (n=354) | High |

Continued

**Table 1** Continued

| Author, country | Study design and setting | Participant characteristics | Health outcomes (measurement tools) | Key findings | QA scores |
|---|---|---|---|---|---|
| Joshi et al,[5] Nepal | Cross-sectional questionnaire survey, Kathmandu (International Airport and nearby hotels/lodges) | Returnee Nepalese male and female migrant workers from Qatar, Saudi Arabia and UAE (n=408) Male=377 (92.4%) Female=31 (7.6%) Mean age (SD)—32 (6.5) years Age ranges—18–53 years | Prevalence of health problems using self-reported/questionnaire survey | ▶ Prevalence of health problem(s)—56.6% (95% CI 51.6 to 61.4) (n=231) ▶ Most common problems: ▶ Headache or fever—30.7% (95% CI 24.8 to 37.1) (n=71) ▶ Respiratory symptoms—21.2% (95% CI 16.1 to 27.0) (n=49) ▶ Musculoskeletal problems—19.9% (95% CI 14.9 to 25.6) (n=46) ▶ Gastrointestinal illness—19.5% (n=45) ▶ Injuries/poisoning—13.9% (95% CI 9.6 to 18.9) (n=32) ▶ Prevalence of some type of injury or accident at their workplace—25% (95% CI 20.8 to 29.5) (n=102) ▶ Health insurance in host countries—36.5% (95% CI 31.8 to 41.4) (n=149) ▶ Sought health services or treatment in the working countries—83.1% (95% CI 42.1 to 51.0) (n=192) ▶ Lack of provision of leave during health problem(s)- 48.7% (n=19) | High |
| **Infectious diseases related issues** | | | | | |
| Al-Awadhi et al,[42] Kuwait | Retrospective analysis of routine healthcare data (2015–2017) | Migrant workers in Kuwait Total examined participants—1000 Nepalese—3.3% (n=33) Age—NR Gender—NR | Prevalence of T solium by screening blood using a sensitive taeniasis-specific anti-rES33 antibody assay | ▶ 6.1% (95% CI 0.7 to 20.0) (n=2) of Nepalese migrant worker sample tested for T Solium taeniasis-specific IgG antibodies | High |
| Sahimin et al,[29] Malaysia | Cross-sectional study (September 2014–August 2015) | Migrant workers from manufacturing, services, agriculture and plantation, construction and domestic work sectors in Malaysia Total participants—610 Nepalese—(n=103) Gender—NR Age—NR | Measure prevalence of E. dispar and E. histolytica | ▶ E. dispar 4.9% (95% CI 1.4 to 12.2) and E. histolytica infections 3.7% 95% CI 0.8 to 10.4 | High |

Continued

**Table 1** Continued

| Author, country | Study design and setting | Participant characteristics | Health outcomes (measurement tools) | Key findings | QA scores |
|---|---|---|---|---|---|
| Sahimin et al,[31] Malaysia | Cross-sectional study | Migrant workers in Malaysia. Total stool samples examined—388 Nepalese—20.9% (81) Gender—NR Age—NR Gender—NR Age – NR | Prevalence of Giardia duodenalis and Cryptosporidium parvum | ▶ Giardia duodenalis 1.8% (0.7–3.7) and Cryptosporidium parvum 0.3 (0.0–1.4), respectively | High |
| Dafalla et al,[41] UAE | Cross-sectional survey conducted at public health clinic | Immigrant workers—food handlers, babysitters, housemaids, drivers working in Sarjaha, UAE Total participants—21 347 (number of Nepalese workers not reported) Total infected population—3.3% (n=708) Gender—NR Age—NR | Prevalence of parasitic infections (examined microscopically and screened for intestinal parasites) | ▶ Proportion of infected migrant workers that are Nepalese—6.2% (95% CI 4.5 to 8.2) (n=44)<br>▶ All protozoal infections: 7% (95% CI 5.9 to 8.6) (n=33)<br>▶ All helminth infections: 4.2% (95% CI 9.8 to 35.3) (n=9) | Moderate |
| Noordin et al,[26] Malaysia | Cross-sectional survey (September 2014–August 2015) | 484 migrant workers from manufacturing, services, agriculture and plantation, construction and domestic work sectors. Nepalese—21.3% (n=103) Gender—NR Age—NR | Prevalence of parasitic infections | ▶ Sero-prevalence of brugian Lymphatic Filariasis (BmR1)—2.9% (95% CI 0.6 to 8.2) (n=3)<br>▶ Prevalence of parasitic infections (BmSXP)—12.6% (95% CI 6.8 to 20.6) (n=13) | Moderate |

**Table 1** Continued

| Author, country | Study design and setting | Participant characteristics | Health outcomes (measurement tools) | Key findings | QA scores |
|---|---|---|---|---|---|
| Sahimin et al,[27] Malaysia | Correctional survey (September 2014–August 2015) | 484 migrant workers Nepalese respondents—20.5% (n=99) Conducted at five working sectors (manufacturing, construction, plantation, domestic and food services) | Sero-prevalence T.gondii through Questionnaire survey and laboratory blood tests | ▲ Sero-prevalence: ▲ IgG—74.7% (95% CI 65.0 to 82.9) IgM—6.1% (95% CI 2.3 to 12.7) | High |
| Woh et al,[28] Malaysia | Cross-sectional study | Healthy, asymptomatic migrant food handlers. Total participants—317 Nepalese—25.2% (n=80) Gender—NR Age—NR | Prevalence of Salmonella carriers, using stool samples | ▲ Prevalence of salmonella among Nepalese migrant food handlers—3.7% (95% CI 0.7 to 10.5) (n=3) | Moderate |
| Abu-Madi et al,[12] Qatar | Retrospective analysis of routine healthcare data (2005–2014) | Records held at Hamad Medical Corporation data-base for subjects referred for stool examination Total participants—29286 Nepalese—4.8% (n=1429) Gender—NR Age—NR | Proportion of helminth infections positive cases | ▲ Highest proportion of helminth infections among Nepalese workers—15.3% (95% CI 13.39 to 17.12) | High |
| Abu-Madi et al,[13] Qatar | Retrospective analysis of routine healthcare data | Recently arrived migrant workers in Qatar Total participants—2486 Nepalese—15% (n=373) Gender—NR Age—NR | Presence of intestinal parasites (helminths and protozoa) | ▲ Proportion of positive cases in Nepalese migrant workers: ▲ Helminths combined—6.2% (95% CI 3.8 to 9.6) ▲ Hookworms—4.3% (95% CI 2.4 to 7.3) ▲ Protozoa combined—13.7% (95% CI 10.0 to 18.2) | Moderate |
| Humphery et al,[16] Qatar | Community-based survey, Doha | Total participants—126 Nepalese—29.3% (n=37) All male population Median age (IQR) in years=33 (27–39) | Prevalence of gastrointestinal pathogens (detected using PCR) | ▲ Total prevalence of gastrointestinal pathogens=62.7% (95% CI 53.6 to 71.1) (n=79) ▲ Gastrointestinal pathogens among Nepalese migrant workers—26.6% (95% CI 10.6 to 24.3) (n=21) | Moderate |

Continued

**Table 1** Continued

| Author, country | Study design and setting | Participant characteristics | Health outcomes (measurement tools) | Key findings | QA scores |
|---|---|---|---|---|---|
| Woh et al,[30] Malaysia | Cross-sectional survey (October 2014– May 2015) | Migrant food handlers living in Malaysia Total participants—383 Nepalese—24.8% (n=95) Gender—NR Age—NR | Knowledge and practices regarding the food handlings | ▲ Mean knowledge scores on: ▲ Symptom of foodborne illness among Nepalese migrant— M=18.4%, SD=28.8 ▲ Food cleanliness and hygiene—M=73.1%, SD=15.3 ▲ Proportion of food handling practices among Nepalese migrant ▲ Poor practices—21.9% (n=7) ▲ Moderate—14.3% (n=32) ▲ Good—43.8% (n=56) | Moderate |
| Imam et al,[18] Qatar | Retrospective analysis of routine healthcare data (January 2006 and December 2012) | Patients with suspected or confirmed tuberculous meningitis. Total participants—80 Nepalese—37% (n=30) Gender—NR Age—NR | Clinical presentation, diagnosis, treatment, outcome and the incidence of adult tuberculous meningitis | ▲ 30/80 patients with tuberculous meningitis were Nepalese (37.5% (95% CI 26.9 to 49.0)). No further data | High |
| Chattu and Mohammad,[40] Saudi Arabia | Retrospective analysis of routine healthcare data from Qassim region (January 2005–December 2009) | Migrant workers (n=165) Male—42% (n=70) Female—58% (n=95) Age—NR | Proportion of reported TB cases, using laboratory test | ▲ Proportion of migrant workers with TB from Nepal: 7% (95% CI 3.8 to 12.3) (n=12) | Moderate |
| Abu-Madi et al,[23] Qatar | Cross-sectional survey (June–September 2009) | Patients resident in Qatar who were randomly recruited and conducted survey—1538 Nepalese—15.3% (n=236) Gender—Male—98.3% (n=232) Female—1.7% (n=4) Age—mean age 28.2 years | Prevalence of intestinal parasitic infections among food handlers and housemaids | ▲ Prevalence of all types of parasitic infections (species)—29.7% (95% CI 25.51 to 34.15) ▲ Helminths—23.7% (95% CI 19.91 to 27.98) ▲ Hookworms—17.8% (95% CI 14.40 to 21.73) ▲ A. lumbricoides—2.5% (95 CI (1.40 to 4.50) ▲ Prevalence of all Protozoa—9.7% (95% CI 7.23 to 12.93) ▲ B. hominis—3% (95% CI 1.69 to 5.01) ▲ Prevalence of non-pathogenic: ▲ Amoebae—3% (95% CI (0.69 to 5.01) ▲ G. duodenalis—3.4% (95% CI 2.02 to 5.52) | High |

**Table 1** Continued

| Author, country | Study design and setting | Participant characteristics | Health outcomes (measurement tools) | Key findings | QA scores |
|---|---|---|---|---|---|
| Ibrahim et al,[17] Qatar | Community based survey, Alkhor hospital | Anti-HEV IgG Nepalese migrants nationally—86 58 of these seen at Alkhor Hospital. Gender—NR Aged 26.7 (SD-5.6, range 19–41 years) | Prevalence of hepatitis E (using ELISA test) and other clinical symptoms | ▶ Prevalence of acute HEV among those seen at Alkhor Hospital—74% (95% CI 60.9 to 84.7) (n=43) admitted to hospital—95.3% (95% CI 84.1 to 99.4) (n=41) | Moderate |
| Chan et al,[24] Malaysia | Cross-sectional survey conducted in a plantation and detention camp of Malaysia | Total foreign migrant workers—501 Nepalese—5% (n=26) Gender—NR Age—NR | *Toxoplasma gondii* IgG and IgM sero-prevalence | ▶ Prevalence of *Toxoplasma gondii* IgG—46.2% (95% CI 26.5 to 66.6) (n=12) ▶ Prevalence of *Toxoplasma gondii* IgM—11.5% (95% CI 2.4 to 30.0) (n=3) | High |
| Al-Marri,[14] Qatar | Population-based retrospective analysis (January 1996–December 1998) | Total cases of positive *M. tuberculosis* culture and sensitivity—406 Nepalese migrant cases—11% (n=44) Gender—NR Age—NR | Drug resistant cases of TB (where positive isolates identified) | ▶ Of total 386 cases of pulmonary TB (321 expats) identified, 11% (95% CI 7.9 to 14.2) n=44, Nepalese cases of TB, of which 9 cases were drug resistant | High |

ELISA, enzyme-linked immunosorbent assay; GCC, Gulf Cooperation Council; HEV, Hepatitis E Virus; NR, not reported.

(four and below). Qualitative studies were assessed by using the JBI qualitative study critical appraisal tool. The checklist contains ten items with domains covering methodological approach, data collection, analysis and interpretation, researcher's role, participants' voice and ethics. The studies were rated high quality (eight and above), moderate quality (between five and seven) or low quality (four and below) as on the previous publication.[11] The assessment was undertaken independently by two reviewers (SPW and KK) with any discrepancies resolved by a third reviewer (PP). As the number of studies in this population is limited, we did not exclude studies based on quality assessment. The results of the quality assessment are presented in online supplemental appendix 2.

### Patient and public involvement

This review was conducted as a part of a project to develop a culturally relevant intervention to support the health and wellbeing of Nepalese migrant workers in GCC countries. Migrants workers were involved throughout the project duration, including the formulation of research question for this systematic review.

## RESULTS

### Screening results

Database searches yielded 2770 articles. After duplicate removal, titles of the 2562 articles were screened and 2253 were excluded. Abstracts of the remaining 309 publications were further screened and 215 of these were excluded. Full text screening of the remaining 94 papers were carried out and a further 61 papers were excluded for various reasons (figure 1). Altogether, 33 papers were included in this review; 31 were quantitative and two were qualitative studies.

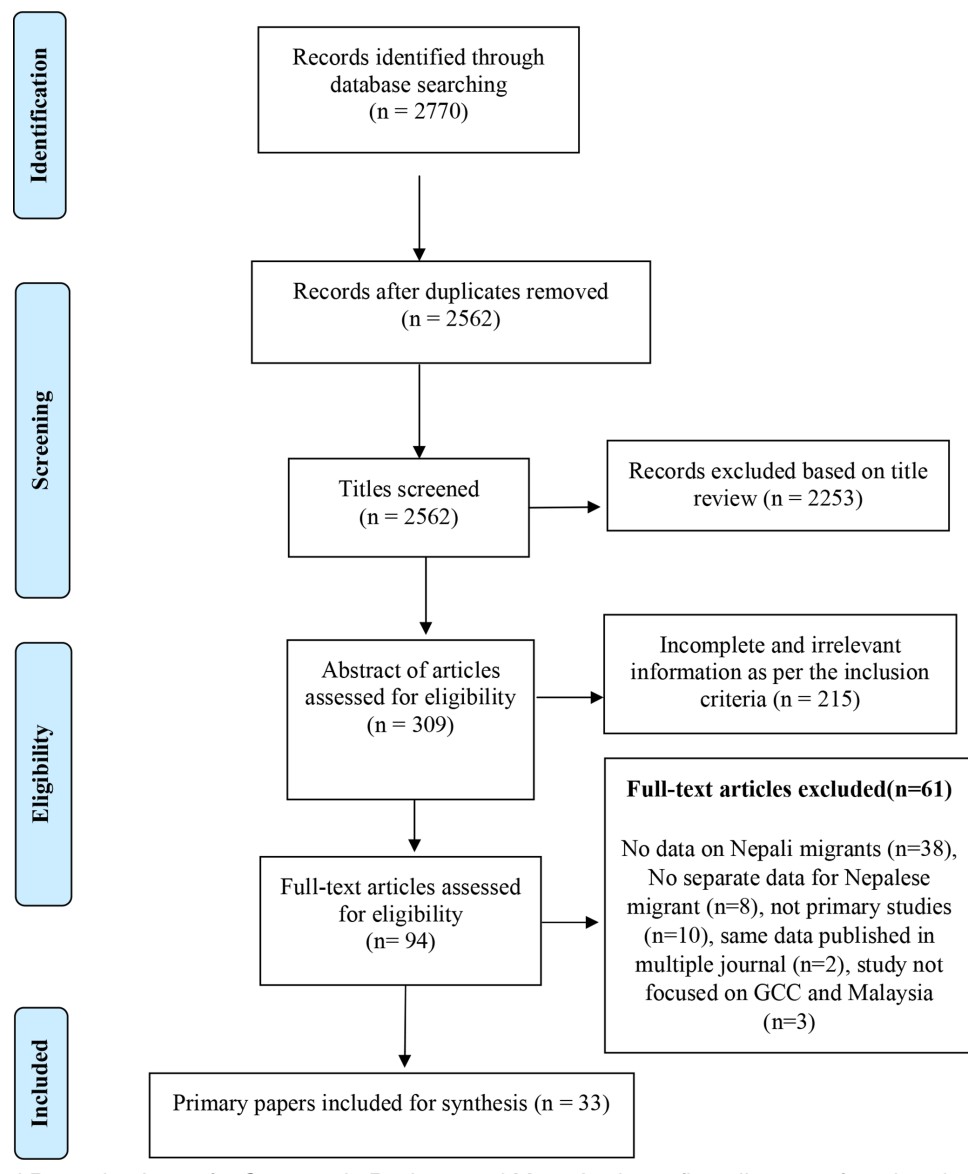

**Figure 1** Preferred Reporting Items for Systematic Reviews and Meta-Analyses flow diagram of study selection. GCC, Gulf Cooperation Council.

## Study characteristics

A total of 33 papers were included in the review among them 12 studies were conducted in Qatar,[12–23] 8 in Malaysia,[24–31] 9 in Nepal,[5 7 32–38] 2 in Saudi Arabia,[39 40] 1 each in UAE[41] and Kuwait,[42] respectively. Two study included all GCC countries and Malaysia,[7 38] another two study included Malaysia, Qatar and Saudi Arabia[32 33] and further two included in Qatar, Saudi Arabia and UAE[5 34] (table 1). The study design varied across the studies; the review included 13 retrospective analysis of routine health-care data[7 12–15 18 20 25 35 37 39 40 42] and 18 cross-sectional studies.[5 16 17 19 21–24 26–34 41] Only two studies were qualitative in nature.[36 38] Nine studies focused specifically on Nepalese migrants as their primary study population[5 7 32–38] while the remaining 24 studies mentioned Nepalese migrant workers as part of a sub-analysis (table 1). In majority of the studies, there was a lack of disaggregated data on demographic characteristics of Nepalese migrant workers. There was a paucity of research with female migrant workers, with just one study identified in this review.[7] The study mainly fell into two categories: those exploring the health risks and experiences of migrants while abroad and those focusing on infectious diseases (mostly done as a part of arrival screening).

## Studies exploring health risk and experiences

### Occupational health and hazards

Seven studies (four high quality and three moderate quality) specifically assessed occupational morbidity, mortality and fitness to work in the destination countries.[15 20 25 32 35 36 39] Majority of these studies were conducted in male migrant workers and the sample of Nepalese migrants varied from 20 to 38 908. Adhikary *et al* reported that around one-fifth (17%) of migrant workers had experienced work related accidents.[32] Poor working environment and not being registered with a doctor was associated with a greater perceived health risk at the work place. Another study reported that over a quarter (27.9%) of migrant workers had experienced occupational injuries: more than half (52%) of these workers fell from a height, 20% had injuries due to fall of a heavy object, 17% had motor vehicle accident injuries, 5% had machinery injuries and remaining 5% had other work related injuries.[15] In a study conducted in Saudi Arabia, Nepalese migrant workers were the third-most unfit population to work; 1.6% were unfit due to the presence of infectious disease and 5.3% due to non-communicable disease.[39] Another study reported that more than one-quarter (25.4%) of migrants had traffic related pedestrian injuries during abroad work[20] (table 1). A study by Pradhan *et al*[35] conducted a retrospective analysis of Government of Nepal data from 2009–2017 and recorded 1345 deaths, of which workplace accident and road traffic accidents contributed to 12% and 10% deaths, respectively. 33 cases of work-related ocular injuries were reported in one study among Nepalese patients of the 440 patients attending a hospital in Malaysia.[25] One qualitative study explored workplace accidents in GCC and Malaysia and reported

several issues faced by the workers including lack of workplace safety, long working hours resulting in dehydration, heat stroke, injuries and accidents-related issues including life-long disability.[36]

### Sexual health

Only one moderate quality study in this review assessed the knowledge, attitudes and perceptions (KAP) of HIV/AIDS related risks.[34] The study was conducted among 408 adult Nepalese migrants (92% male) with at least 6 months of work experience in one of the three Gulf countries (Qatar, Saudi Arabia and UAE). The study showed that 90% of respondents had concerns about HIV/AIDS, and 17.2% of workers reported having sexual intercourse with a partner other than their spouse within the last 12 months. More than half (59%) of the respondents perceived themselves at high risk of being infected due to their sexual activities[34] (table 1).

### Mental health

Five studies (all moderate quality) examined mental health issues among migrant workers. The sample of Nepalese migrants workers in these studies ranged between 20 and 1354.[7 21 33 35 38] One study on Nepalese female returnee migrant workers from Middle East and Malaysia reported the prevalence of mental health problems as 8.3%.[7] Another study reported that almost a quarter (23%) of labour migrants to Malaysia, Qatar and Saudi Arabia had experienced mental health issues, with a strong positive link between perceived health risk in the work environment and mental health status.[33] Third study reported a paradoxical finding with 4% increase in the predicted probability of depressive symptoms among Nepalese migrant workers compared with Arab, for every unit increase in perceived quality of life.[21] One study analysed Nepalese government's report and looked at 1354 deaths in Nepalese migrant workers, of which 8.5% were due to suicide.[35] The fifth quality qualitative study reported various mental health problems among the workers including loneliness, social isolation, tensions, anxiety, attempt to suicide[38] (table 1).

### Healthcare access

Five studies (one high and four moderate quality) focused on labour migrants' healthcare access issues and the number of Nepalese workers in these studies ranged between 20 and 942, respectively.[5 7 32 37 38] Adhikary *et al*[32] reported that workers who were not registered with a doctor had poor health outcomes compared with those who were registered. Another study also reported that only 36.5% workers had access to health insurance and about half (48.7%) did not have paid sick leave during their health problems.[5] Another study on Nepalese female returnee migrant workers reported that only 11% of respondents received health services during their abroad work.[7] The fourth study reported that only insurance 68% of the workers had health insurance abroad and only 20% underwent regular health check-up.[37] In the qualitative

study, participants reported poor access to mental health services related, mainly related to communication problems, and stigma to mental health[38] (table 1).

### Other health issues

A total of five studies (two high and three moderate quality) involving participants number ranging between 44 and 1354 reported various health issues.[7 19 22 35 37] One study on Nepalese female returnee migrant workers reported a prevalence of workplace abuse, torture or maltreatment at the workplace, and physical harm at 37%, 31% and 11%, respectively.[7] Clinical prevalence of oral lesions among migrant workers was found to be 4.6%.[19] Third study looked at the chronic kidney disease among workers and found that 13.6% of workers had diabetic nephropathy.[37] In the study by Pradhan et al,[35] cardiovascular disease, natural/others reasons and murder contributed to 42%, 25% and 1.7% of deaths, respectively. The last study reported that of patients attending to the emergency medical service in Qatar, out-of-hospital cardiac arrest among Nepalese migrant patients was found to be 11.6%[22] (table 1).

### Studies on infectious diseases (parasitic and bacterial infections, tuberculosis and hepatitis E)

Of the 33 included studies, 17 studies (nine high and eight moderate quality) reported the proportion of sero-and/or faeco positive cases of infectious diseases (parasitic and bacterial gastroenteric infections, tuberculosis (TB), hepatitis E).[12–14 16 17 19 23 24 26–31 40–42] The number of Nepalese workers included in these studies ranged between 12 and 1429. In several of these studies, Nepalese migrant workers had the higher proportion of infectious disease cases among the population studied. These infectious diseases included, toxoplasmosis (46.2%, working in Malaysia),[24] TB (7%, Saudi Arabia and 11%, Qatar),[14 40] TB meningitis (37.5%, Qatar),[18] diarrhoeal bacterial infection (26.6%, Qatar),[16] protozoan ova/cysts (13.7%), helminths (6.2%) and hookworms (4.3%, Qatar),[12] hepatitis E (74%, Qatar),[17] Brugian Lymphatic Filariasis (BmR1) (2.9%, Malaysia) and parasitic infection (BmSXP) (13%, Malaysia).[26] Moreover, prevalence of salmonella among Nepalese migrant food handlers (3.7% Malaysia),[28] mean knowledge of food cleanliness and hygiene (73.1%, Malaysia) and symptom of foodborne illness (18.4% Malaysia)[30] (table 1).

### Overall quality assessment

More than half of the cross-sectional prevalence studies (54% n=15/28) scored as 'high' quality and remaining were of moderate quality.[7 12 15–17 19 26 28 30 35 37 40 41] Similarly, three analytical studies were rated as moderate quality[21 32 33] and the two qualitative studies were rated as one high and one of moderate quality.[36 38] None of the studies were rated as poor quality. The results of the quality assessment scores are presented in table 1 and details are presented in online supplemental appendix 2.

### DISCUSSION

To our knowledge, this is the most comprehensive review of the health and well-being status/issues of the Nepalese migrant workers in the GCC countries and Malaysia. The resultant lack of disaggregated demographic data means that the overall characteristics of Nepalese participants is difficult to determine. The dissonance between issues covered in the peer-reviewed and grey literature for this population, namely in national and international media and in government reports, is notable. Disproportionately few studies focused on occupational mental, and sexual health of migrant workers.

### Occupational health

Our review identified seven papers focusing on occupational morbidity, mortality and fitness to work in the destination countries.[15 20 32 35 36 39] Only three of these focused solely on Nepalese migrants, and none compared occupation or working conditions with morbidity and mortality experienced.[32 35 36] This a crucial gap in the literature and further studies are needed to guide policy change. There has been widespread media coverage of the poor working conditions faced by Nepalese migrant workers and health impacts of these conditions are highlighted by the plight of manual labourers working for the forthcoming 2022 FIFA Qatar World Cup. Close to a fifth of labour migrants to Malaysia, Qatar and Saudi Arabia had experienced a workplace accident.[32] According to a Nepalese government report, there were circa 7467 deaths among Nepalese migrant workers abroad between 2008/2009 and 2018/2019, and over 40% of the deaths were deemed either of natural or other/unidentified cause.[3] Despite these workers being young (mean age 29 years) and fit (assessed by health screening both at home and destination countries), the magnitude of the proportion of these deaths is unusual in these groups.[3] This raises questions about robustness of postmortem investigative practices and classification methodologies, a concern highlighted by both the Nepalese government and civil society groups.[43] Indeed, Pradhan et al[35] suggest that many deaths attributed to cardiovascular diseases and 'natural causes' correlate with longer hours worked in high temperatures in this setting. It is worth noting that Nepalese migrant workers themselves are not oblivious to these occupational risks—those who reported a poor or very poor work environment were found to be 3.5 times more likely to suffer a workplace accident.[32]

### Mental health

Five studies in the review reported on mental health issues. Adhikary et al[33] reported that almost a quarter of labour migrants to Malaysia, Qatar and Saudi Arabia had experienced mental health issues, with a strong positive correlation between perceived health risk in the work environment and mental health status. The qualitative study by Regmi et al[38] highlighted various mental health problems among the workers including loneliness, anxiety and attempt to suicide. Similar findings

were reported in a cross-sectional study of 5000 migrant workers in Shanghai, where 21% reported mental disorders such as obsessive-compulsive disorder, anxiety, and hostility.[44]

The Nepalese government report suggests that suicide is a significant cause of mortality in labour migrants to GCC countries and Malaysia, and there is evidence that mental health is an underexplored issue facing this population.[45–47] Only one of the study in this review looked at the suicide cases with nearly 10% of the deaths in these workers resulting from suicide.[35] The paucity of peer-reviewed studies exploring risk factors of poor mental status and psychiatric morbidity for this population requires urgent attention.

Migration for work is a time of significant turmoil: new language, new culture and poor working conditions. Loss of protective familial and wider social networks exacerbate feelings of homesickness, loneliness and hopelessness that commonly develop among this population.[48–50] Psychiatric under-diagnosis is common in deprived populations and is compounded by poor screening of those with pre-existing psychiatric conditions.[51–54] The result is lack of mental health support and omission of medications in destination contexts that can worsen conditions. Most common psychiatric morbidity in this population centred around depressive and anxiety-related disorders, although the impact of addiction particularly of alcohol consumption remains underexplored.[47 55–57] The impacts of labour migration on the mental health of left-behind families is also important, but beyond the scope of this review.[45 58]

### Sexual health

Only a single study in this review examined sexual health issues among this population and exploring HIV/AIDS KAP among Nepalese migrant workers. Joshi et al[34] reported that over 17% had had sexual intercourse with someone other than their spouse or partner during the final 12 months of their stay abroad. This highlights higher levels of sexual risk taking behaviour, echoed by studies focusing on Nepalese migrants to India, which showed widespread use of local female sex-workers by male Nepalese migrant populations, multiple sexual partners and low levels of condom use. While there may be differences between the Indian and GCC or Malaysian contexts, the authors note there is a clear dearth of evidence around non-HIV/AIDS related sexual health of these migrants, and the impact of this on left-behind families.[59 60] Similar findings also revealed from the studies in Bangladesh and China among migrant workers at high risk of heterosexual HIV acquisition.[61 62]

### Infectious disease

Out of 33 studies, 17 studies focused on migrant workers in a destination country and provided minimal disaggregated analysis on the Nepalese sub-population. Majority of these were done as a part of arrival screening and focus on infectious diseases were conducted from a destination country perspective. Overwhelmingly, the discussion sections of these studies focused on Nepalese migrant workers as potential vectors for transmitting infectious diseases to native population. This health security framing overlooks Nepalese labour migrants as a vulnerable population by virtue of their poor socioeconomic status in their origin country as well poor working and living conditions, and poor access to healthcare in destination countries.[5 63 64] Similar findings were also reported in a study from Singapore where a relatively high prevalence of malaria, hepatitis and TB was reported among migrant workers in Singapore.[65] Migrant workers in South Asia generally appear to have a greater prevalence of infectious diseases due to the complex interaction of several factors—this includes higher prevalence of infectious diseases in their native countries together with aforementioned poor access to healthcare and low socioeconomic status.[6] Acknowledgement and consequent introduction of policies to improve these structural drivers of infectious diseases among Nepalese migrants would be a more holistic approach that might both better protect the local population and improve the health and well-being of the vulnerable migrant population.[66]

### Literature gap for female migrant workers

Women comprise only 8.5% of Nepalese labour migrant abroad.[3] However, the role of women in the migration story is far more significant and complex than this figure betrays with regards to true numbers of women migrating, roles of women 'left behind' and how it has influenced gender norms in Nepalese society. The complex interplay between various factors such as sociocultural norms, women's role in decision-making and freedom to mobility reflect on their health from access to sexual and reproductive health services to gender-based violence.[67] Just one study has previously attempted to capture health outcomes among female migrants.[7] They highlighted that almost a quarter of female Nepalese migrants faced multiple health problems and over 37% had faced workplace abuse, with close to half of the 3% that reported becoming pregnant while away doing so as a result of sexual abuse.[7]

Female labour migration from Nepal has increased significantly over the past decade, driven by increasing demands in primarily GCC destination countries, poor agricultural employment opportunities and a slowly changing gender norms.[68] One third of remittances to Nepal are from female migrant workers.[7 69] Higher proportion (90%) of female labour migrants are undocumented workers in Gulf countries and this may have resulted from the restrictive governmental labour migration policies such as prohibition of women to work in the Gulf domestic sector.[70] Precarious channels of migration bring greater risks of exploitation and harm to health,[71] yet neither the peer-reviewed literature in health, nor do wider literatures reflect the magnitude of these issues. More work is required on the health of Nepalese female

migrants abroad, as well the challenges in reintegration that they face on their return.[68]

## Strengths and limitations

This review has several strengths. As mentioned earlier, the review is the most comprehensive review to date on this population. As GCC and Malaysia are the most attractive destinations for migration, the findings of this review will have important research implications in terms of highlighting the research gap on specific health problems of migrant workers in general as well as the lack of research focus on female migrant workers. This review also has important practical implications, such as informing the design of culturally appropriate care and outreach for Nepalese workers. Studies were not restricted based on particular health outcomes, peer-reviewed studies looking at a range of health issues in this population were included. Screening of studies and quality assessment was conducted by two independent reviewers, ensuring low risk of selection bias in this review. We applied research design specific quality assessment tools, providing the accurate ratings of the articles. However, there were a number of limitations. The review did not systematically include grey literature although a number of key reports were used as reference points to compare to our findings from the peer-reviewed literature. The risk of missed studies by only searching English language databases is noted, particularly through exclusion of relevant Nepalese peer-reviewed journals. Also, recent guidelines have been published on reporting of narrative synthesis without meta-analysis,[72] however, these guidelines are more applicable for intervention studies, thus we have not used these in this narrative systematic review. As the number of qualitative studies were very small (n=2), we reported the key findings from these studies rather than conducting a separate meta-synthesis.

## CONCLUSION

This review identified a number of health issues among Nepalese migrant workers in the GCC countries and Malaysia, namely those centred on occupational, mental and sexual health of migrants, and infectious disease, together with health-related issues facing female labour migrants. While there are early signs that Nepal may be moving beyond its predominantly remittance economy, there is no doubt that labour migration to Malaysia and the GCC countries is the reality facing an entire generation of working age Nepalese. The studies identified by the review highlight the need for improved health support, whether through regular health checks in destination countries, more stringent policies and legislation around permissible working conditions or better preparation for migration through more relevant pre-departure training. The findings suggest the urgent need to progressive policy changes, both in Nepal and destination countries, to better protect the health of labour migrants

and improve their access to essential health services and acceptable working conditions.

**Contributors** PP and JAC designed and supervised the study. PP wrote the review protocol, conducted the literature search and wrote the final draft of the manuscript. SPW and KK screened the articles, extracted the data, carried out quality assessment and contributed to the initial drafts. PP, JAC and AM obtained funding for the study. JAC, AM and PS reviewed and edited the manuscript. All authors read and approved the final manuscript.

**Funding** This review was funded from Research England's institutional allocation from the Global Challenges Research Fund (Reference Number G2626).

**Competing interests** None declared.

**Patient consent for publication** Not required.

**Provenance and peer review** Not commissioned; externally peer reviewed.

**Data availability statement** No primary data were generated for this study.

**ORCID iDs**
Priyamvada Paudyal http://orcid.org/0000-0002-6209-575X
Jackie A Cassell http://orcid.org/0000-0003-0777-0385
Anjum Memon http://orcid.org/0000-0001-8256-301

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
