## [Reviewer comments · BMJ Open]

ARTICLE DETAILS

TITLE (PROVISIONAL)	Health and Wellbeing Issues of Nepalese Migrant Workers in the Gulf Cooperation Council Countries and Malaysia: A Systematic Review
AUTHORS	Paudyal, Priyamvada; Kulasabanathan, Kavian; Cassell, Jackie; Memon, Anjum; Simkhada, Padam; Wasti, Sharada

VERSION 1 – REVIEW

REVIEWER	Nicola Pocock London School of Hygiene and Tropical Medicine, UK
REVIEW RETURNED	06-May-2020

GENERAL COMMENTS	Thank you (all authors) for writing a very succinct paper that summarizes the main health risks and outcomes among Nepali migrant workers. This is an important subject especially given the deaths we are seeing among this population with no known cause which you refer to in this article. I offer suggestions for how to improve the review below. Methods (Recommend in future that the authors use PRISMA-P format to construct the protocol – some information is missing in the current protocol) p.5 – did you use keywords only, or MESH terms as well? Please specify (in Appendix 1 it is currently unclear). p.5 – you will need to justify why EMBASE and MEDLINE only were searched. EMBASE is European focussed while MEDLINE is N. America. Why was Global Health, or Scopus, not included when they have much better LMIC coverage? p.5 – were qualitative studies eligible? Your use of JBI prevalence critical appraisal tool suggests not, but please specify exact study designs that were eligible for inclusion. This has implications for how you report/justify use of JBI prevalence tool only (please see below). p.6 - Multiple studies using same data source – explanation for large sample size criteria? It is fine to include multiple articles from the same study/data source, as long as both included N's are reported (Item 6 in PRISMA explanation article). p.6 - Was the data extraction form piloted? Please specify.
---

	p.6 – Data extraction and synthesis – please specify that meta-analysis was not undertaken and why (typically authors report heterogeneity of outcomes/methods to measure outcomes) p.6 – Were only prevalence studies identified in the review? The JBI also has critical appraisal tools for analytical cross-sectional studies, cohort studies, etc. Please include a justification for using the prevalence critical appraisal tool (CAT) only. p.6 –Narrative synthesis incurs risk of selective reporting of outcomes in tables. PRISMA extension in this 2019 Campbell article offers guidance on how to conduct narrative synthesis and reporting. I have only learned of this guideline recently myself. Please clarify whether/if these were followed - or state that it was not followed in the limitations section, which may incur a risk of bias. p.6 – in Quality Assessment, please specify what domains the CAT covered (e.g. aspects of sampling, statistical analysis methods) as per good practice. For scoring, please specify that 9 items are included in the prevalence QAT total. In Liberati 2009 PRISMA elaboration article, Box 4. distinguishes between study quality and risk of bias. The former is the best the authors could do in the setting, vs. theoretical grounds for risk of bias (RCTs fare better than quasi-experiments in the hierarchy of evidence). I think you are focusing on quality of studies, rather than theoretical risk of bias in your assessments - please specify somewhere in this section, referencing the PRISMA elaboration article. An example is in Quality appraisal section of this review. Results p.6 – Screening results – do you mean that 2 further studies were identified from backwards citation tracking of the included 21 studies? (not a hand search of journals?) Please specify. Study characteristics p.8 – 2 cohort studies were included (re. point about selecting prevalence CAT only, above) Overall, quality ratings should be threaded throughout the discussion, e.g. “One medium rated study found that...” – you could specify the total number of low, medium, high quality studies per sub-section in the first sentence. It is helpful to have a sense of where the higher quality evidence lies per topic. Discussion p.10 – Sexual health should be mentioned here (alongside occupational and mental health). p.11 - Mental health - in reference to poor screening practices and underdiagnosis, you can consider citing this paper Mental Health Assessments section (the quote referred to Nepali workers, although that is not specified directly in the paper). p.13, line 3 – ‘All these’ – please specify what ‘these’ are p.13, line 7 – ‘We commend Simkhada et al for their efforts...’ – potential conflict of interest as Simkhada is a co-author on the paper - suggest rephrasing this, e.g. ‘Just one study has
--	--

	previously attempted to capture health outcomes among female migrants' p.13, line 38 – Please suggest briefly what the important research and practical implications are (e.g. for research on specific health problems, to inform service delivery/culturally competent care for Nepali workers, etc) line 45 – methodological robustness – having two independent reviewers alone does not guarantee overall robustness in a systematic review. Suggest to replace with 'ensuring low risk of selection bias in this review'. A further limitation is not applying research design specific CATs (your use of prevalence CAT only), which may have provided more accurate ratings. Appendix p.19 – Table 1 title, include N of studies in brackets (N=23) p.26 – PRISMA flow diagram – ideally you would provide the N for each reason in Full text exclusions – optional suggestion - you would score higher in AMSTAR-2 by doing this. p.32 PRISMA checklist – I did not see any reference to selective reporting of outcomes within studies, or publication bias on p.6 – suggest that you assess this by taking a look at positive vs. null results reported (depends if you were only looking at prevalence? This is currently unclear – Table 1. reports ORs, suggesting analytical studies were included)
--	--

REVIEWER	Alison Reid Curtin University, Western Australia
REVIEW RETURNED	07-May-2020

GENERAL COMMENTS	Nepalese migrants review Comments for the Editor Thanks for the opportunity to review this manuscript. It is an interesting topic about which very little has been written. I've listed a few things below that I think need more clarification but on the whole I think it is a well written paper. Comments for the Author Thanks for the opportunity to review this manuscript which systematically reviews the health issues of Nepali migrant workers to GCC countries and Malaysia. Overall the paper is well written and the methods appear sound. However, I don't think you have drawn enough out of the study results and have listed some specific areas for clarification below. Page 8 – study characteristics. Can you clarify what a prospective observational study is – do you mean case-control or cohort study (or other type of observational study?) An observational study is not a study type alone – it is a description of a group of studies in epidemiology – observational versus experimental. Throughout your description of the study results, you have not provided enough detail about each study you are describing. For example, sentence one occupational health hazards you state that four studies examined ohs. However, you then don't tell the reader anything about those studies, other than the findings. For example, how many participants were included in the study, how many reported the finding you presented, what other
--

	characteristics or migration information is relevant for this study. Specifically more context is needed when you discuss the findings of each study. This was a consistent point throughout the results. Also, many percentages are presented but no measure of distribution? Can you put 95% CIs around those percentages (can you create your own if they are not in the original papers) Page nine – why is infectious disease a different heading style to the others? First paragraph of the discussion – there is a reference to lack of disaggregated data – this is the first reference to this? Can you make include something about this in the results? Occupaitonal Health paragraph of the discussion – what should the p% of deaths be from natural or unidentified causes. You state what it is – but you also need to state what it should be – to highlight that it is a problem. You could also emphasise that, in the main, migrant workers are young and fit – so death is unusual in these groups. Female labour migration – you include a paragraph in the discussion about this – but it is not raised elsewhere. I think you could include a section in the results about it – to at least highlight the paucity of research in the area.
--	---

VERSION 1 – AUTHOR RESPONSE

Reviewers' comments	Response to the comments
Reviewer 1	
(Recommend in future that the authors use PRISMA-P format to construct the protocol – some information is missing in the current protocol)	Thank you very much for your advice, we will use the PRISMA-P format in the future.
p.5 – did you use keywords only, or MESH terms as well? Please specify (in Appendix 1 it is currently unclear).	Only Keywords were used, this has now been clarified in the appendix 1.
p.5 – you will need to justify why EMBASE and MEDLINE only were searched. EMBASE is European focussed while MEDLINE is N. America. Why was Global Health, or Scopus, not included when they have much better LMIC coverage?	Thank you for your suggestion, we have now included both Global Health and Scopus in addition to EMBASE and MEDLINE databases. The new search was carried out on the 8th of May 2020 and has identified some new articles (including the recent articles published after our last search in May 2019). The PRISMA diagram has been updated and the manuscript has been revised accordingly to include the results of the newly identified studies. (please see line 99-108 in the methods section and revised appendix 1)

p.5 – were qualitative studies eligible? Your use of JBI prevalence critical appraisal tool suggests not, but please specify exact study designs that were eligible for inclusion. This has implications for how you report/justify use of JBI prevalence tool only (please see below).	As we had not limited any study design in our search, all study designs were eligible. We did not identify any eligible qualitative studies in our old search but two qualitative studies were identified in the search carried out in 8th of May. Both of these studies were published recently and these have been included in our revised draft. (line 164-165)
p.6 - Multiple studies using same data source – explanation for large sample size criteria? It is fine to include multiple articles from the same study/data source, as long as both included N's are reported (Item 6 in PRISMA explanation article).	We included the studies with the longest follow-up data available. For e.g Abu-Madi published three articles using data from medical records from 2005 to 2008, again from 2005-2011 and final one from 2005-2014. So we included the last one as this included all the samples from the earlier studies.
p.6 - Was the data extraction form piloted? Please specify	Yes, the data extraction form was piloted for three studies (Al-Thani et al 2015, Dafalla 2017 and Chattu 2017).
p.6 – Data extraction and synthesis – please specify that meta-analysis was not undertaken and why (typically authors report heterogeneity of outcomes/methods to measure outcomes)	As there was heterogeneity in the outcome measured and the measurement tools used in the studies, we were unable to conduct meta-analysis. This limitation has been reported in the discussion section. (line 414-415)
p.6 – Were only prevalence studies identified in the review? The JBI also has critical appraisal tools for analytical cross-sectional studies, cohort studies, etc. Please include a justification for using the prevalence critical appraisal tool (CAT) only.	The JBI CAT tool has been revised as suggested; although the two included studies reported the design of their studies as 'prospective', both studies by Humphrey 2016 and Ibrahim 2009 were cross-sectional studies. In the revised draft, prevalence JBI tool has been used for all studies describing prevalence estimates and analytical JBI tool has been used for studies reporting effect sizes. Qualitative JBI tool has been used for the newly identified qualitative studies in the revised draft. (Please see Appendix 2)
p.6 –Narrative synthesis incurs risk of selective reporting of outcomes in tables. PRISMA extension in this 2019 Campbell article offers guidance on how to conduct narrative synthesis and reporting. I have only learned of this guideline recently myself. Please clarify whether/if these were followed - or state that it was not followed in the limitations section, which may incur a risk of bias.	The limitation has been acknowledged in the discussion section. (line 416-418)
p.6 – in Quality Assessment, please specify what domains the CAT covered (e.g. aspects of sampling, statistical analysis methods) as per good practice. For scoring, please specify that 9 items are included in the prevalence QAT total.	Thank you, this information has been now added in the revised article in the methods section. (line 145-153)

In Liberati 2009 PRISMA elaboration article, Box 4. distinguishes between study quality and risk of bias. The former is the best the authors could do in the setting, vs. theoretical grounds for risk of bias (RCTs fare better than quasi-experiments in the hierarchy of evidence). I think you are focusing on quality of studies, rather than theoretical risk of bias in your assessments - please specify somewhere in this section, referencing the PRISMA elaboration article. An example is in Quality appraisal section of this review.	As suggested, this has been clarified in the 'Quality Assessment' section in the methodology of the revised manuscript, and the PRISMA article has now been cited. (line 134-137)
Results	
p.6 – Screening results – do you mean that 2 further studies were identified from backwards citation tracking of the included 21 studies? (not a hand search of journals?) Please specify.	Yes, two articles were identified from backward citation of the included articles. These articles, however, were identified in the new electronic search carried on the 8 May, without the need for backward citation. This has been reflected in the revised PRISMA diagram. (please see fig 1)
Study characteristics p.8 – 2 cohort studies were included (re. point about selecting prevalence CAT only, above) Overall, quality ratings should be threaded throughout the discussion, e.g. “One medium rated study found that...” – you could specify the total number of low, medium, high quality studies per sub-section in the first sentence. It is helpful to have a sense of where the higher quality evidence lies per topic.	As suggested, quality ratings have been threaded throughout the result section. (line 186-273)
Discussion	
p.10 – Sexual health should be mentioned here (alongside occupational and mental health)	Thank you, we have added this now. (line 293)
p.11 - Mental health - in reference to poor screening practices and underdiagnosis, you can consider citing this paper Mental Health Assessments section (the quote referred to Nepali workers, although that is not specified directly in the paper).	We have now cited this paper on the discussion section. (please see ref 56)
p.13, line 3 – ‘All these’ – please specify what ‘these’ are	We have now explained ‘these factors’ (line 381-383)
p.13, line 7 – ‘We commend Simkhada et al for their efforts...’ – potential conflict of interest as Simkhada is a co-author on the paper - suggest rephrasing this, e.g. ‘Just one study has previously attempted to capture health outcomes among female migrants’	We have amended the sentence accordingly. (line 384)

p.13, line 38 – Please suggest briefly what the important research and practical implications are (e.g. for research on specific health problems, to inform service delivery/culturally competent care for Nepali workers, etc)	We have added this information in the discussion section. (401-405)
line 45 – methodological robustness – having two independent reviewers alone does not guarantee overall robustness in a systematic review. Suggest to replace with ‘ensuring low risk of selection bias in this review’. A further limitation is not applying research design specific CATs (your use of prevalence CAT only), which may have provided more accurate ratings.	This has been amended in the revised draft. We have now added the design specific CAT as a strength of this review. (line 408-409)
Appendix	
p.19 – Table 1 title, include N of studies in brackets (N=23)	This information has been added in the draft. (line 283)
p.26 – PRISMA flow diagram – ideally you would provide the N for each reason in Full text exclusions – optional suggestion - you would score higher in AMSTAR-2 by doing this.	This information has been included in the revised PRISMA diagram (fig 1)
p.32 PRISMA checklist – I did not see any reference to selective reporting of outcomes within studies, or publication bias on p.6 – suggest that you assess this by taking a look at positive vs. null results reported (depends if you were only looking at prevalence? This is currently unclear – Table 1. reports ORs, suggesting analytical studies were included)	Studies were predominately cross-sectional reporting the prevalence. Hence it was not possible to make overall assessment about selective reporting of outcomes.

Reviewer 2	
Can you clarify what a prospective observational study is – do you mean case-control or cohort study (or other type of observational study?) An observational study is not a study type alone – it is a description of a group of studies in epidemiology – observational versus experimental.	As mentioned above, although the two included studies reported the design of their as ‘prospective’, both studies (Humphrey 2016 and Ibrahim 2009) were cross-sectional studies. We have now changed the terminology accordingly in our manuscript. (please see table 1)
Throughout your description of the study results, you have not provided enough detail about each study you are describing. For example, sentence one occupational health hazards you state that four studies examined ohs. However, you then don’t tell the reader anything about those studies, other than the findings. For example, how many participants were included in the study, how many reported the finding you presented, what other characteristics or	Thank you very much for your comment. Although it is not possible to comment on every single studies in the texts (as this information has been presented in the table), we have threaded the information about studies in the description as much as possible in the revised draft. (please see the result section (line 186-273))

migration information is relevant for this study. Specifically more context is needed when you discuss the findings of each study. This was a consistent point throughout the results.	
Also, many percentages are presented but no measure of distribution? Can you put 95% CIs around those percentages (can you create your own if they are not in the original papers	We have now presented the results with 95% CI. (please see table 1)
Page nine – why is infectious disease a different heading style to the others?	There were two types of studies; those experiencing health risks and experience (page 8) and those done as a part of arrival screening (infectious diseases, page 9). The different style headings are used for these two types of studies.
First paragraph of the discussion – there is a reference to lack of disaggregated data – this is the first reference to this? Can you make include something about this in the results?	We have included this information in the results section in the revised draft. (line 179-180)
Occupaitonal Health paragraph of the discussion – what should the p% of deaths be from natural or unidentified causes. You state what it is – but you also need to state what it should be – to highlight that it is a problem. You could also emphasise that, in the main, migrant workers are young and fit – so death is unusual in these groups	According to a study by ILO on Nepalese migrants' death (http://ilo.org/wcmsp5/groups/public/---asia/---ro-bangkok/---ilo-kathmandu/documents/publication/wcms_493777.pdf), “unidentified cause and natural cause are too ambiguous to provide any useful information for developing interventions to ensure the wellbeing of migrant workers. As well, it is not known at this point how the different destination countries categorize mortality and morbidity”. The report also highlights that the deaths under these categories do not indicate the cause of death specifically and is epidemiologically unhelpful. We are not aware of any data on the proportion of expected deaths in migrant workers due to unidentified cause and natural cause, so we are unable to comment on that. As advised, the information about unusual deaths is stated in the discussion section. (line 306-309)
Female labour migration – you include a paragraph in the discussion about this – but it is not raised elsewhere. I think you could include a section in the results about it – to at least highlight the paucity of research in the area.	We have highlighted the paucity of research in the result section. (line 180-182)

VERSION 2 – REVIEW

REVIEWER	Nicola Pocock London School of Hygiene and Tropical Medicine, UK
REVIEW RETURNED	14-Jul-2020

GENERAL COMMENTS	Thank you to the authors for revising this important review, I look forward to sharing this in my network once it is published. Please find some minor outstanding suggestions below: Line 25 – here and in line 111, suggest to remove > and replace with text for ease of reading/flow ‘...migrant workers aged 18 or older working in...’ Line 29 – As you now use study specific JBI tools, rephrase to ‘All included studies were critically appraised using Joanna Briggs Institute study specific tools.’ Line 30 - please remove the link to the protocol (looks out of place in the Abstract), suggest to replace with ‘A review protocol was developed and registered on the University of Sussex website’ Line 137 onwards – when referring to the JBI tools, avoid starting sentence with ‘Checklist for...’ – suggest to rephrase - ‘The JBI prevalence study critical appraisal tool was used...’ For all of the tools you mention. Line 181 – rephrase – ‘there was a paucity of research with female migrant workers, with just one study identified in this review’ Line 403-405 suggest to rephrase - ‘This review also has important practical implications, such as informing the design of culturally appropriate care and outreach for Nepalese workers.’ Optional suggestion depending on authors preference - As there was heterogeneity in the outcome measured and the measurement tools used in the studies, we were unable to conduct meta-analysis. This limitation has been reported in the discussion section. (line 414-415) - this is not a limitation of your review per se, but just needs to be stated in your Methods > Data extraction and synthesis section. I am flagging just so the authors are aware that it is not their limitation (it is a well conducted review, and all other limitations have been mentioned in the discussion).
---

REVIEWER	Alison Reid Curtin University, Australia
REVIEW RETURNED	06-Jul-2020

GENERAL COMMENTS	All of my earlier comments have been addressed by the authors
---

VERSION 2 – AUTHOR RESPONSE

Reviewers' comments	Response to the comments
Reviewer 1 Please state any competing interests or state 'None declared': None declared	This has been stated at the end of the draft (line 456).
Line 25 – here and in line 111, suggest to remove > and replace with text for ease of reading/flow '...migrant workers aged 18 or older working in....'	The sentence has been amended as suggested (line 29-30).
Line 29 – As you now use study specific JBI tools, rephrase to 'All included studies were critically appraised using Joanna Briggs Institute study specific tools.'	This has been amended in the revised draft (line 34-35).
Line 30 - please remove the link to the protocol (looks out of place in the Abstract), suggest to replace with 'A review protocol was developed and registered on the University of Sussex website	The link has been removed and sentence has been amended as suggested (line 50-51).
Line 137 onwards – when referring to the JBI tools, avoid starting sentence with 'Checklist for....' – suggest to rephrase - 'The JBI prevalence study critical appraisal tool was used....' For all of the tools you mention.	All the sentences have been amended accordingly (line 151-163).
Line 181 – rephrase – 'there was a paucity of research with female migrant workers, with just one study identified in this review'	This sentence has been rephrased (line 193-194).
Line 403-405 suggest to rephrase - 'This review also has important practical implications, such as informing the design of culturally appropriate care and outreach for Nepalese workers.'	The sentence has been rephrased as suggested (line 415-417).
Optional suggestion depending on authors preference - As there was heterogeneity in the outcome measured and the measurement tools used in the studies, we were unable to conduct meta-analysis. This limitation has been reported in the discussion section. (line 414-415) - this is not a limitation of your review per se, but just needs to be stated in your Methods > Data extraction and synthesis section. I am flagging just so the authors are aware that it is not their limitation (it	Thank you, this is mentioned in the methods section (line 142-144) and we have now removed the sentence from the discussion section as suggested.

is a well conducted review, and all other limitations have been mentioned in the discussion).	
---	--